

# Changes in the Surface Salinity Gradient and Transport of the Irminger Current: The Climate Perspective

Nathan Paldor[1], Ofer Shamir[1,2], Andreas Münchow[3], and Albert D. Kirwan Jr.[3]

[1]Fredy and Nadine Herrmann Institute of Earth Sciences, Hebrew University of Jerusalem, Jerusalem, Israel
[2]Present Affiliation: Courant Institute of Mathematical Sciences, New York University, NYC NY, USA
[3]School of Marine Science and Policy, University of Delaware, Newark DE, USA

**Correspondence:** Nathan Paldor (nathan.paldor@huji.ac.il)

**Abstract.**

Here we use a new analysis schema, the Freshening Length, to study the transport in the Irminger Current on the east and west sides of Greenland. The Freshening Length schema relates the transports on either side of Greenland to the corresponding surface salinity gradients by analyzing climatological data from a data assimilating global ocean model. Surprisingly, the

warm and salty waters of the Current are clearly identified by a salinity maximum that varies nearly linearly with distance along the Current's axis. Our analysis of the climatological salinity data based on the Freshening Length schema shows that only about 20% of the transport east of Greenland navigates the southern tip of Greenland to enter the Labrador Sea in the west. The other 80% disperses into the ambient ocean. This independent quantitative estimate based on a 37-year long record complements seasonal to annual field campaigns that studied the connection between the seas east and west of Greenland

more synoptically. A temperature-salinity analysis shows that the Irminger Current east of Greenland is characterized by a compensating isopycnal exchange of temperature and salinity, while west of Greenland the horizontal convergence of less dense surface water is accompanied by downwelling/subduction.

## 1 Introduction

The flow around Cape Farewell at the southern tip of Greenland is a critical component of the complex circulation of the At-

lantic Meridional Overturning Circulation known as AMOC (Marshall and Schott, 1999; Bryden et al., 2020). This is illustrated in Figure 1, adapted from Drinkwater et al. (2020) and Little et al. (2019), which shows two recent sketches of the circulation of the subpolar North Atlantic. A comparison between the two panels underscores the complexity of the three-dimensional subpolar circulation: The red and dark blue curves represent warm and cold surface currents, respectively, while light blue represent deeper flows that result from deep convection or deep overflow from Denmark Strait and the Faroe Channel, both of

which connect the subpolar Atlantic to the Greenland Sea north of Iceland.

The spatial arrangement of the warm and salty Irminger Current adjacent to a cold and fresh East Greenland Current that together form a southward flowing western boundary current (Våge et al., 2011) is particularly noteworthy. The East Greenland Current advects polar waters from the Arctic Ocean along the continental slope region to Denmark Strait (Håvik et al., 2017) and beyond (Pickart et al., 2005). Furthermore, a buoyancy-driven coastal current forms from local coastal freshwater dis-

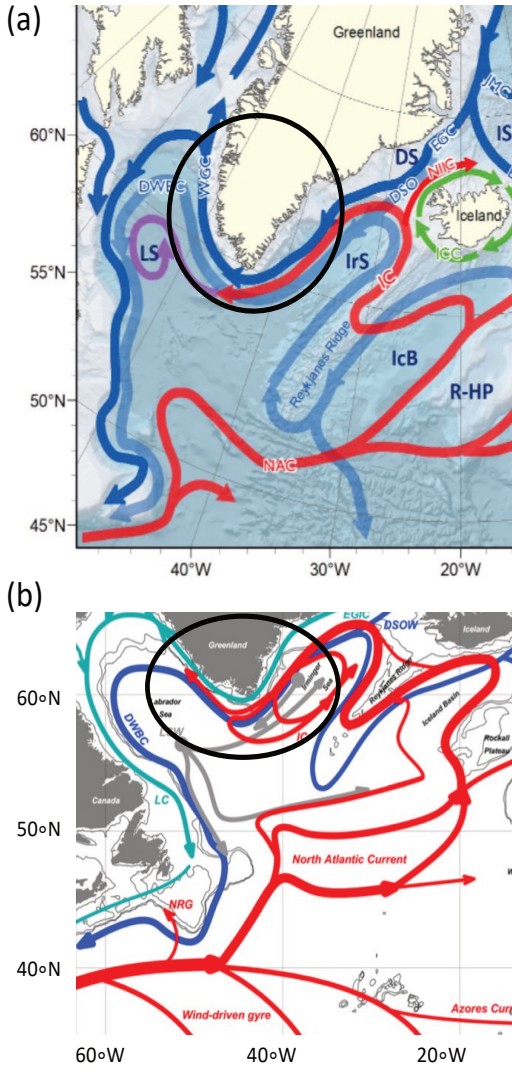

**Figure 1.** Two sketches of the North-Atlantic circulation between Iceland in the north-east and Newfoundland in the south-west adapted from (a) Drinkwater et al. (2020) (in polar-stereographic projection) and (b) Little et al. (2019) (in Mercator projection). Red currents indicate warm and salty surface waters. Light blue currents in (a) and blue currents in (b) indicate cooler deeper flows. The dark blue lines in (a) and turquoise lines in (b) indicate cold and fresh Arctic surface waters such as the East Greenland and Labrador Currents to the east of Greenland and Labrador, respectively. Grey dots and arrows in (b) indicate deep water formation areas in the Labrador and Irminger Seas. Black circles focus on the Irminger Current at the southern tip of Greenland, the region of interest in this report.



charges as described by Bacon et al. (2002) and Sutherland and Pickart (2008). These southward currents all converge as they approach Cape Farewell. Also of interest are the southward currents off East Greenland, which appear to continue northward off West Greenland in Figure 1b only. Most of these waters turn cyclonically near 65 N latitude, however, a fraction extends northward into Baffin Bay as a prominent Atlantic temperature maximum that can be traced beyond 78 N latitude (Münchow, 2016; Rignot et al., 2021). The western limb of the Irminger Current contains an unknown fraction of the eastern limb (Myers

et al., 2009).

It appears then that the transport of salty and warm Atlantic water of subtropical origin plays an important role in the AMOC. As is evident from figure 1 the Irminger Current is an important component of this climate system as it is a conduit for relatively warm and salty waters to mix with colder and fresher waters. Panel a of figure 1 suggests that this current blends into the Labrador Sea and does not flow past Cape Farewell. In contrast, panel b of this figure shows that some fraction of the

Irminger Current continues past Cape Farewell as the West Greenland Current, (Pacini et al., 2020) The latter view is supported by Holliday et al. (2007) who estimated that about 5.1 Sv or 80 % of the Irminger Current retroflects back into the Irminger Sea.

The large number of observational studies conducted in the Irminger and Labrador Seas off southern Greenland inform the kinematics of subpolar boundary currents, deep convection, and climate impacts, (Pickart et al., 2003; Holliday et al., 2007;

Myers et al., 2007, 2009; Le Bras et al., 2020). These studies describe in considerable detail the seasonal and interannual variability of the hydrography and currents from moorings and hydrographic expeditions along cross-sections of the currents, but except for Holliday et al. (2007) generally provide limited information of synoptic transport variations of the Irminger Current east and west of Cape Farewell.

In order to provide additional insight on synoptic variations of the Irminger Current we use nearly 37 years of observations

obtained from a data assimilating global model (Carton et al., 2018). Consequently our salinity data set is dynamically consistent with the velocity fields. From this we are able to quantify the fraction of the Irminger Current to the east of Greenland that successfully negotiates the sharp turn at Cape Farewell, and which contributes to the West Greenland and Baffin Island Currents (Münchow et al., 2015); and eventually the Labrador Current. As was concluded by Pickart et al. (2003) some of the deep Labrador Sea Water must originate in the Irminger Sea. The West Greenland and Irminger Currents are the main conduits

for the transport of this water. While much of the Irminger Sea water detaches from the boundaries as it flows into the Labrador Sea near Cape Farewell, we emphasize that an unknown fraction of the Irminger Current continues to flow northward along West Greenland (Münchow et al., 2015).

To quantify the fraction of the Irminger Current that flows into the Labrador Sea and remains at its surface, i.e. does not subduct, we appeal to the new paradigm of Evaporation Length developed by Berman et al. (2019). That approach utilizes the

relative changes in the salinity, due to the net evaporation that a column of water undergoes along a trajectory, to calculate a new parameter, the Evaporation Length. The method was applied to salinity and geochemical data from the Red and Mediterranean Seas, the Indian Ocean, and the Gulf Stream (Berman et al., 2019). The Evaporation Length estimates the (hypothetical) distance that a moving column of surface water has to travel in order to evaporate all its water. As was shown in (Berman





et al., 2019) the Evaporation Length reliably quantifies the volume transport per unit length of the cross-stream direction of the
current.

We extend this paradigm to the case where the salinity of the water in the column decreases along the trajectory, and in our
application the length parameter is termed Freshening Length. In the scenario considered here the Freshening Length estimates
the (hypothetical) distance that the column has to travel in order for its water to become fresh (i.e. zero salinity). Save for a
trivial change of sign the subsequent quantification of the volume transport (per unit length of the cross-stream direction) of
the current in the freshening scenario is identical to that in the evaporation scenario. The paradigm is particularly suitable for
use in connection with changes in routinely observed variables (salinity in our case) stored in all climatological data archives
since it is measured in all observations and reported in all model calculations. We also use the surface temperature data of this
reanalysis data archive to analyze changes in the properties of surface Temperature-Salinity (T-S) diagrams along the current
to examine the exchange that takes place in the currents east and west of Greenland.

Our approach is based on long-term observations. This differs from previous studies in that the analysis follows the mean
flow and is not limited to brief snapshots in time and not constrained to a specific geographical region. Furthermore, the
new metric is insensitive to interannual variations associated with large decadal freshening of the subpolar North Atlantic as
described most recently by Holliday et al. (2020). Thus, this climatological data base is appropriate for analysis using the
Freshening Length, as it encapsulates long-term trends in the along-flow characteristics of salinity and temperature in this
region.

The next section describes the data used in the analysis. This is followed by the results of our analysis. We conclude with a
synopsis of our findings and a discussion of new issues raised by this research.

## 2    Data and Methods

We use data from the "Simple Ocean Data Assimilation" or SODA project. The technical details of these data are given in
Carton et al. (2018) and the data are freely accessed at https://rda.ucar.edu/datasets/ds650.0/. The spatial resolution of the
gridded data assimilation product is 0.5 degrees in latitude and longitude while the temporal resolution is 5 days. Time series
of salinity and temperature data span the period from January 3, 1980 to December 19, 2017, nearly 37 years. The surface
salinity and temperature values presented here are averages over the entire 37-year record taken at a depth of about 5 m. The
geographical region used here is 45°W and 35°W longitude between 55°N and 65°N latitude. It covers the ocean adjacent to
the southern part of Greenland.

In the first step of the analysis we determine, at each latitude, the longitudes of the maximum salinity. Then we identify the
maximum salinity along the axis of the Irminger Current by calculating a 5-point zonal average (90 km) centered on the local
salinity maximum. From this we calculate transects to the east and west of Cape Farewell (Nunap Isua in Greenlandic), which
is the southern tip of Greenland near 44°W longitude and 60°N latitude. The distance along the transect, $x$, was calculated as
the spherical geodesic distance along the trajectory formed by the points of maximal salinity values. Hence $\Delta x$ is the geodesic



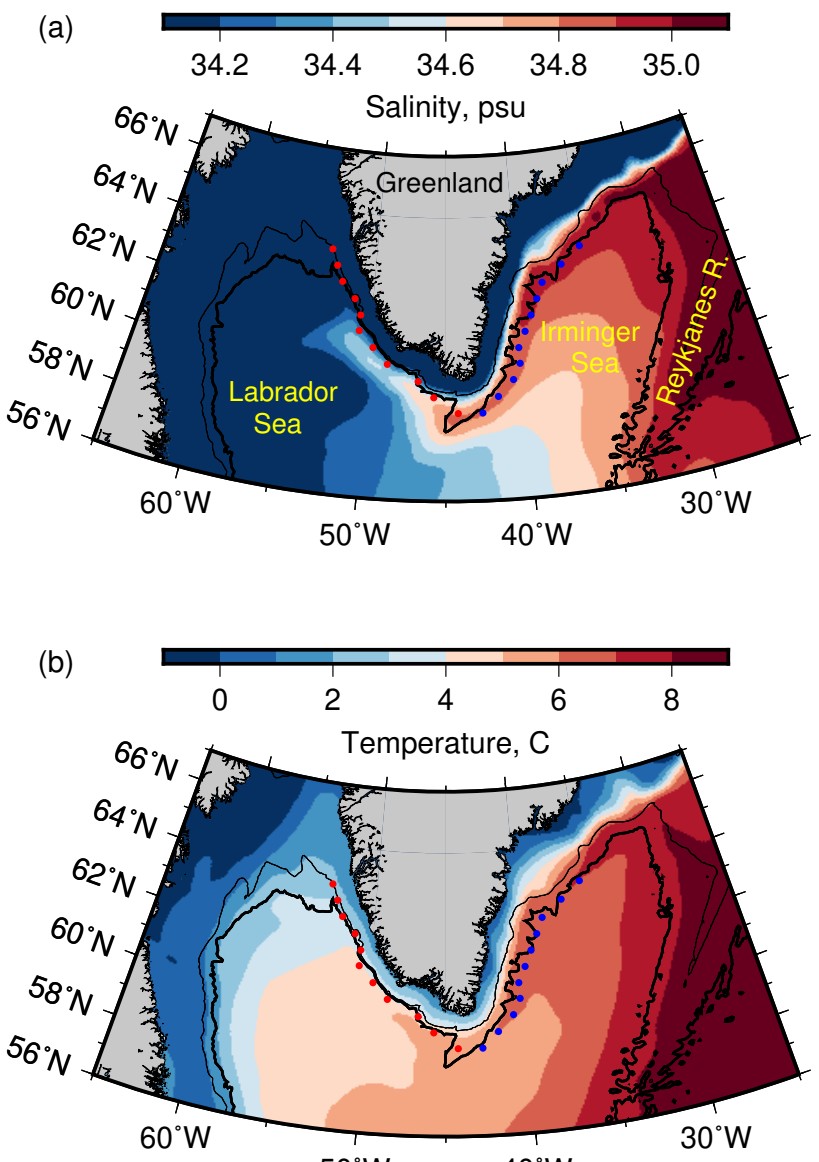

**Figure 2.** 37-year averages of surface salinity in psu (a) and temperature in °C (b) around southern Greenland. Blue and red filled circles indicate the location of eastern and western salinity maxima, respectively, along constant longitudes. Black contours are 1000 m and 2000 m bottom depths.

distance between two adjacent salinity maxima. Figure 2 shows the location of the zonal salinity maxima to the east (in blue symbols) and west (in red symbols) between Cape Farewell and 63.75° N.





## 3 Results

Figure 2 shows the 37-year mean surface distribution of salinity (panel a) and temperature (panel b) between the Reykjanes

Ridge in the east and Labrador in the west. The Reykjanes Ridge to the south of Iceland is part of the Mid-Atlantic Ridge system where salinities and temperatures generally exceed 35 psu and 8 °C, respectively, at the surface. The climatological mean clearly suggests a cyclonic circulation in the Irminger Sea that distributes warm and salty Irminger surface water preferentially near the 2000-m isobath, which is consistent with modern mooring observations (de Jong et al., 2012). This figure also shows that much fresher water occupies the continental shelf and slope regions off Greenland delineated by the 1000-m isobath to the

east of 55° W longitude. The surface water of the Irminger Current wraps around Cape Farewell and extends northward into the Labrador Sea. Along its path, however, it becomes fresher and colder as it encounters a polar water mass, which we discuss next.

Different mixing regimes are indicated by the T-S diagrams of surface waters on both sides of Greenland shown in Figure 3. The data from the eastern (Irminger Current) limb (shown by blue triangles) all fall closely on a density contour of 1027.4 Kg·

m$^{-3}$. In contrast, the data from the western (West Greenland Current) limb (shown by red triangles) follow an almost straight line that crosses density contours between 1027.4 and 1026.6 Kg ·m$^{-3}$ (for location of points see Figure 2). While the T-S diagram clearly indicates that different mixing processes act in the two limbs, it provides no quantitative information on the rate at which the salty water of the Irminger Current freshens as it entrains fresh water along its flow. In order to determine the rate we now introduce an extension of the Evaporation Length paradigm.

Berman et al. (2019) showed that the inverse of relative salinity changes due to net evaporation along a transect, $S_0/\frac{\Delta S}{\Delta X}$, (where $S_0$ is the salinity at the origin of the trajectory and $\frac{\Delta S}{\Delta X}$ is the salinity gradient along the trajectory) yields the value of the Evaporation Length denoted by $L$. The Evaporation Length schema focuses on a water column of depth $H$ that moves at the ocean surface with speed $U$ and whose salinity change is determined by $q$, the rate at which fresh water is removed (say, by net evaporation) while its salt content remains unchanged. The conservation of mass of salt and water in the moving column

relates $L$ and $q$ to the column's depth, $H$, and speed, $U$, via the linear relation:

$$qL = HU = F \tag{1}$$

where $F$ (=$HU$) is the volume transport per unit length in the cross-stream direction.

In the present study we extend the original Evaporation Length paradigm to the freshening scenario where the salinity decreases along the trajectory (i.e. $\Delta S < 0$) since fresh water is added to the column e.g. by entrainment. In both cases the

salinity change of the water in a column that extends from the surface to a depth $z = -H$ yields a length that estimates either the (hypothetical) distance that the column has to travel for all its water to evaporate (in the evaporation paradigm) or for its water to loose all its salinity and become fresh water (in the freshening paradigm).

The Freshening Length paradigm is applied here to the Irminger Current that bifurcates from the salty North Atlantic Current. This current flows northward along the western flank of the Reykjanes Ridge, turns cyclonically as does the 2000-m isobath,

and flows southward along the slope of southeast Greenland (DeJong et al., 2020). Since it is surrounded by fresher waters on both the Greenland's shelf and the Irminger Sea sides (Figure 2), it is clearly identifiable by a local salinity maximum. The




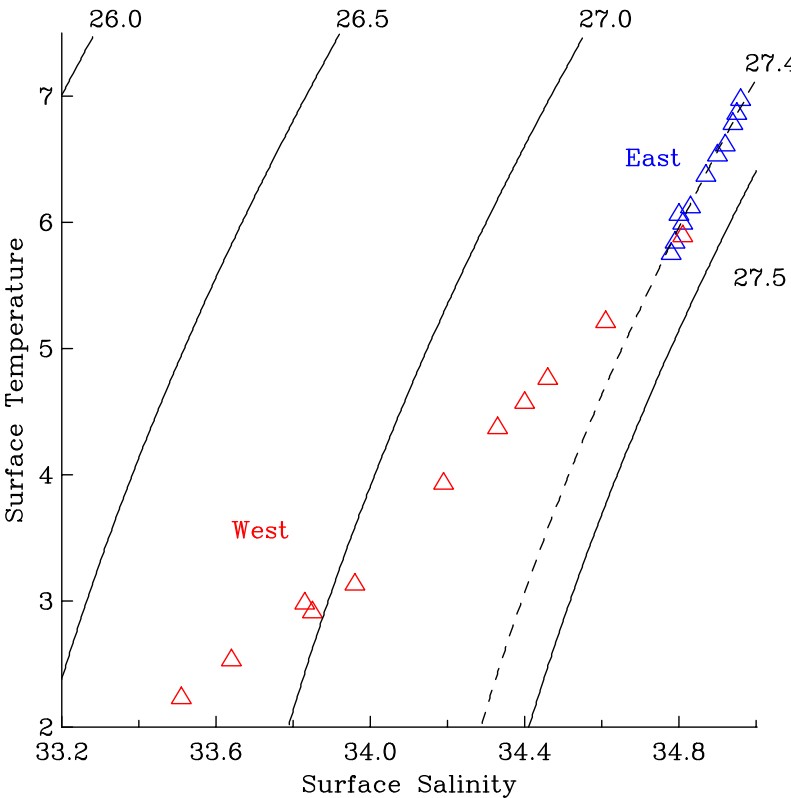

**Figure 3.** Surface temperature-salinity diagram (triangular symbols) over density contours (solid and dashed thin lines) for the stations shown in Figure 2 and used in Figure 4. Red and blue triangles denote stations along the western and eastern limbs, respectively.

paradigm is also applied to the West Greenland Current, which is also marked by a local salinity maximum and that flows northwestward east of Greenland.

Figure 4 shows the salinity variations along the Irminger current (panel a) and the West Greenland Current (panel b) along with their best fitting least-squared linear approximations. The curves in the two panels are continued from the right bottom corner of panel (b) to the upper left corner of panel (a). Figure 4 clearly shows that the Freshening Length, $L$, on Greenland's east side ($1 \times 10^5$ Km, panel b) is about five times larger than on the west side ($0.2 \times 10^5$ Km, panel a). The correlated variance $R^2$ of salinity and distance along the transect to the east and west off Greenland are 0.92 and 0.97, respectively. Notice that the eastern transect starts at 63.75N and extends to the south-west, while the western transect starts at 58.75N and extends to the north-west to the same 63.75N latitude.

We find a 5-fold decrease in the value of $L$ from the eastern to the western limbs of the Irminger Current. According to Equation (1) this change in $L$ results from a combination of changes in the current transport, $F$, and the rate of fresh water entrainment, $q$ which is discussed in the next section.




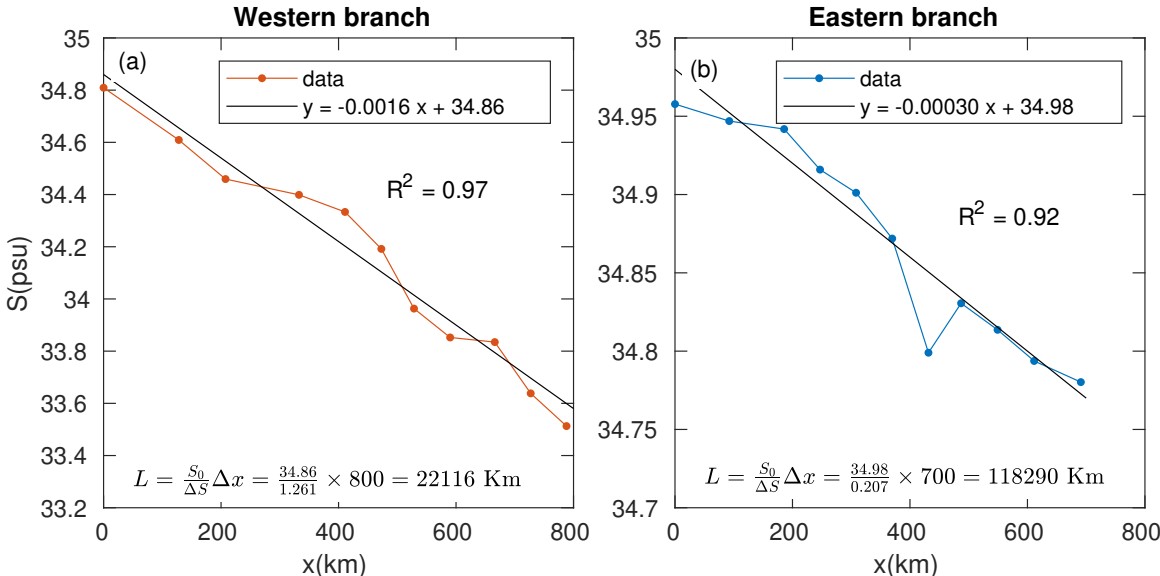

**Figure 4.** Salinity (psu) along the trajectories highlighted in figure 2. The salinity in each limb is averaged over the 5 grid points containing the maximum and two adjacent grid points on either sides of the maximum. $x$ is the spherical geodesic distance along the trajectory, i.e. $\Delta x$ is the geodesic distance between two adjacent maxima. The linear fits were calculated using Matlab curve-fitting tool to find the least square linear fit.

## 4 Summary and Conclusions

In view of the tremendous impact that AMOC has on Earth's climate, it important to clarify which of the flows proposed in the two panels of 1 is correct. To answer this question we applied a new analysis technique, the Freshening Length paradigm (Berman et al., 2019), to 37 years of assimilated modeled hydrographic data. As noted above, this paradigm compliments traditional observational studies that utilize data from moorings and surveys of the Irminger and the West Greenland Currents.

Consider first the T-S plot in figure 3. Note that the southernmost stations on the east and west Greenland have nearly

identical T-S values, even though they are over 75 km apart. This implies that the West Greenland Current is a continuation of the Irminger Current. Figure 3 also indicates different freshening mechanisms of the Irminger Current east of Greenland and the West Greenland Current. The Irminger Current exhibits typical isopycnal mixing from the northern start to the southern end, a distance of about 700 kilometers. We interpret this isopycnal mixing as horizontal entrainment along a density surface combined with release of heat to the atmosphere. This conclusion is consistent with both (Pickart et al., 2003) and Smith

(1937). While the former employed modern moorings and cross-sectional hydrography, early American (Smith et al., 1937) and Danish (Riis-Carstensen, 1936) studies provided similar descriptions as part of the International Ice Patrol. As opposed to these quasi-synoptic survey and seasonal mooring data, we here used climatological data that average properties over 37 years.





In contrast to the Irminger Current, the Western Greenland Current is characterized by much more intense cooling and freshening. The intense cooling and freshening west of Greenland cannot result from the localized and slow release of heat to

the atmosphere accompanied by slow entrainment of surrounding water as on the eastern Irminger Current. Instead, it probably results from horizontal convergence of cold and fresh water accompanied by downwelling (subduction of the current). This, too, is consistent with Pickart et al. (2003) who concluded that the source of the North Atlantic Deep Water originates in the Irminger Sea rather than the Labrador Sea. A comparison between the T-S properties in the two limbs of the Irminger Current shows that the slight freshening along the eastern limb is density-compensated by cooling. In contrast, the more substantial

freshening by polar waters along the western limb is not density-compensated by cooling, which result in a decrease of the water density there. The distinctive difference of the mixing processes to the east and west of Greenland resolves the discrepancy in circulations in Figure 1. This analysis suggests that only about 20% of the Irminger Current continues northwestward as the West Greenland Current.

Our analysis of the SODA3 climatological salinity transects along the Irminger and West Greenland Currents employs the

Freshening Length paradigm recently proposed in Berman et al. (2019). The $L$-value of $0.2 \times 10^5$ km calculated here for the West Greenland Current typifies intensive salinity changes similar to those in the Red Sea while the value of $1 \times 10^5$ km found in the Irminger Current east of Greenland typifies moderate salinity changes such as those in the Mediterranean Sea. For the 5-fold difference in $L$ between the two Currents, equation (1) establishes a single relation between 2 variables – $q$ and $F(= HU)$. The difference between the values of $L$ can be explained by a combination of two extreme cases.

In the first extreme case we assume that the fresh water entrainment rates, $q$, into the Irminger and West Greenland Currents are equal. According to Eq. (1) the fact that $q$ is constant implies that the equatorward transport, $F$, of the Irminger Current is 5-times larger than the poleward transport of the West Greenland Current. In this uniform $q$ extreme case the 80% decrease in $L$ from $1 \times 10^5$ km in the east to $0.2 \times 10^5$ km in the west implies that 80% of the transport of the Irminger Current detaches from it and does not negotiate the sharp turn at Cape Farewell. Some of the 80% may cool, sink (i.e. subduct) or contribute

to the NADW as suggested by the purple cyclone labeled "LS" in Figure 1a. Alternatively, some of these 80% could retroflect and form the southern branch of the Irminger Gyre as depicted in 1b and labeled "IC". Holliday et al. (2007) quantifies such a retroflection from a single snapshot of velocity observations, however, they find that 80% navigates the bend of Cape Farewell from east to west while 20% moves offshore and to the east. This suggests that a large fraction of the 80% is shed as eddies in the area (Bracco et al., 2008).

In the second extreme case, the volume transport, $F$, is assumed constant, which implies that the fresh water entrainment rate, $q$, is 5 times smaller in the West Greenland Current compared to the Irminger Current. This extreme case requires that the speed of the current decreases in the West Greenland Current, to allow for the higher (temporal) rate of entrainment, while the depth of the current increases to maintain the same volume transport.

In addition to the 37-year mean that yields the Freshening Lengths discussed above, the SODA3 data contain annual and

seasonal variations of surface salinity. We repeated the above calculations of $L$ based on annual data and found that the root-mean-square variations of the Freshening Length in the annual data are only about 300 km, i.e. of order 1% or less of the climatological values (not shown). The annual variations during the 37-year period include physical signals such as the Great





Salinity Anomalies in 1981 and 1993 (see Belkin, 2004) that propagate through the subpolar North Atlantic Ocean with surface salinities, $S(x = 0)$, that are more than 0.4 psu below the SODA3 average both east and west of Greenland. We also found

(not shown) that during 1995 and 2016 the salinity along the axis of the Currents east of Greenland were nearly uniform which implies infinitely large Freshening Lengths. The 2016 findings are consistent with those of Holliday et al. (2020) who found substantial expansion of the Irminger Gyre in recent years. The implications of the comparison between climagtological and annual values imply that the Freshening Length paradigm yields a robust metric largely insensitive to decadal events like the Great Salinity Anomalies only when it is calculated from climatological data.

Though the surface salinity data used here is dynamically consistent with SODA3's velocity data, the Freshening Length estimate of the transport is much more robust and informative than direct estimates based on velocity profiles. The reason is that even though cross sections of along-stream velocity in the two Currents yield similar maximal velocities of about 0.2 m·s$^{-1}$, the vertical and horizontal extents of the currents are not accurately determined. Consequently, velocity profiles cannot yield a reliable estimate of the transports in the Currents.

Our result that only 20% of Irminger Current waters navigate around Cape Farewell could be tested by placing cross-sectional current mooring and hydrographic surveys east and west of Greenland. We hypothesize that the 5-fold difference in the value of $L$ in the Irminger and West Greenland Currents is due to changes in volume transport and entrainment rate. This could also be combined with an effort to estimate changes of $L$ directly from concurrent drifter deployments that would also reveal along-track salinity and temperature variations.

*Author contributions.* NP conceived the basic idea of the project, wrote the first draft and took part in all subsequent revisions. OS calculated the salinity transects and Freshening Lengths and produced initial versions of the maps. AM produced the final maps as well as the T-S diagram and participated in writing the manuscript. ADK participated in writing the manuscript

*Competing interests.* The authors declare no competing interests.

*Acknowledgements.* The authors are pleased to acknowledge: We did not receive any support for this research.





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
