# Peer review of "Changes in the Surface Salinity Gradient and Transport of the Irminger Current: The Climate Perspective"

_Ocean Science, 2021_

## Author Comment (AC1)

In this Author Comment #1 the original comments by Reviewer #1 are written in black while the authors' responses are written in blue.

We thank the reviewer for his/her detailed review and for the time s/he spent on it. Nearly all of the comments will be implemented in the revised version, including a change of the main focus from estimation of transport ($F$) when $q$ (fresh water flux which is a proxy for lateral mixing with the surrounding fresher water) is unknown to the estimation of $q$ itself when $F$ is known (from SODA velocity fields). This change of focus results from the direct transport calculations suggested by the reviewer that yielded values of $F$ east and west of Greenland and which have greatly streamlined the sermon of the paper. A discussion of the physical meaning of $q=F/L$ and its application to the two branches of the IC will be added in the revision. This will give the robust determinations of the values of $L$ an easy to grasp physical meaning based on Eq. (1) for known values of $F$ and will focus the sermon of the paper on a quantification of the lateral mixing (entrainment) of the IC east and west of Greenland.

**Summary of review**

This study presents a diagnostic, which is referred to as Freshening Length, to infer along-track changes in the Irminger Current transport around the South-Greenland coast. Based on the Freshening Length, the authors conclude that 20% of the original water mass in the Irminger Current at the eastern side are transported along the southern coast of Greenland to the western side, while 80% are dispersed into the ambient ocean.

Overall, I find this study quite thin. Further analyses may be needed to justify publication. From the current version, I am not able conclude that the study represents a substantial contribution to the journal.

Indeed, our paper has a well-defined focus: The application of the new Freshening Length diagnostic/schema to the Irminger Current. Applying this new schema to climatological reanalyzed sea surface salinity data yields the noted robust and reliable estimate of the changes in $L$ along the Current. No other oceanographic diagnostic yields a similarly confident/reliable estimates. With the new direct transport estimations ($F$) based on SODA's velocity fields (that will be presented in a new table) the application of Eq. (1) to the IC east and west of Greenland will yield quantitative estimates of lateral mixing (entrainment) in the two branches of the IC.

For instance, the manuscript includes four figures and only two of these are original. Figure 1 contains maps adapted from other studies and Figure 2 shows climatological fields of the sea surface temperature and salinity from the publicly available SODA dataset, along with the points at the locations that were used to diagnose the transport changes.

Figure 1 will be eliminated from the revised version and all the figures and the new table (that shows the ratio between the transports east and west of Greenland) will contain original calculations based on SODA's salinity and velocity fields. A new figure (see below) will be added to explain the adaptation of the Evaporation Length schema (which was developed in Berman et al., 2019 and which will be discussed in more

details in the Introduction of the revised version) to the Freshening Length schema employed in this study.

[Figure]

I would recommend publication only if (1) additional analyses are added, or (2) the authors can demonstrate more clearly what the overall value of their analysis is.

Additional analyses could include an exploration of the processes along the current section by which the transport exchanges occur, such as eddies, or further dynamical implications of their study, or temporal changes over or the investigated period and links to larger-scale ocean or atmospheric variability. In any case, I would recommend that the authors further demonstrate the potential information that can be gained from the Freshening Length.

In line with the reviewer's alternative (2), the revised version will demonstrate how the Freshening Length together with direct transport calculations yield a proxy for the rate of horizontal mixing via the value of $q$. Our results imply that the rate of entrainment must change when the IC negotiates Cape Farewell since the transports (calculated from SODA's velocity fields) vary only slightly between the east and west branches of the IC. The application of Eq. (1) will take the form of $q = F/L$ when the RHS is known. The revision will also include a detailed account of the physical meaning of $q$.

**General comments**

1) The scientific writing could be improved. Some paragraphs are difficult to follow. In particular, many paragraphs could be shortened and the sentences could be written in a more concise way. Phrases like *"it appears"* (line 31) do not sound scientific. Overall, I think the amount of text is not in proportion to the amount of information it includes. Therefore, I would recommend shortening of the text.

The text will be shortened by eliminating points that are tangential to the paper's new main sermon.

2) I did not understand why the Freshening Length is important. The climatological map of the sea surface salinity (Figure 2b) already shows there is a gradual freshening along the

current. This is expected since freshwater from the Greenland coastal currents is gradually added along its path. Given that the results are expected already from the climatology map what information is gained from the additional quantification by the Freshening Length?

Other diagnostics, like the freshwater column, which is the integrated freshwater anomaly over depth relative to a pre-defined reference salinity, or the freshwater volume, have been used in numerous earlier studies. These diagnostics have been applied to distinct scientific questions. I am not convinced of the overall value of the Freshening Length unless the authors can demonstrate a clear use or application of the Freshening Length that other diagnostics or just visual inspection of the climatological sea surface salinity map are not able to provide.

The main point in the revised version is that Figure 3 (the T-S diagram) only provides a qualitative comparison between the changes in SSS and SST east and west of Greenland while the values of $L$ yield a quantitative measure for these changes. In the revised version, accurate determination of both L and F yield an estimation of $q$, the parameter that parameterize horizontal mixing with the surrounding fresher water.

3) Based on my understanding of this study, the main result is the statement that 20% of the water in the Irminger Current travel around the southern tip of Greenland. Yet, why is it important to know how much water travels around the southern tip of Greenland as a coherent current? I do not think that the analysis provides meaningful information about the AMOC since the loss of transport that is calculated may be compensated for by other currents and eddies. I cannot see a clear connection to the AMOC from this analysis.

The revised version will focus on the entrainment of surrounding water by the IC on both sides Greenland.

4) The analysis is focused on grid point averages around selected points along the salinity maximum. Yet, the current may be broader at some locations than at others, in which case the diagnostic does not describe the transport in the Irminger Current but is sensitive to how the current is defined. A considerable fraction of the transport could also occur in eddies or in the boundary currents like the Greenland Coastal Currents. These are not captured by focusing on a narrow current with pre-defined width.

True. That's why we emphasize transport per unit length in the cross-stream direction where the current's direction is determined by the local salinity maximum. The new direct transport estimates are also conducted over the same cross-sections (five grid points centered on the point of maximum salinity) so eddies moving independently of the IC are not included in $F$ (but the eddies do not necessarily move parallel to the IC)

5) The analysis only considers the climatological mean over a 37-year period. I do not think there is a substantial gain in such an analysis. It would be more interesting to look at the time variability of the transport and investigate the involved dynamical processes.

The advantage of averaging 37 years of data is that it filters out seasonal, annual and even decadal changes. In the paragraph between lines 184-194 (of the old version) we

note the results of short-period analyses and highlight their consistency with previous studies.

6) Part of the freshwater transport around Greenland's coast occurs as sea ice. I am not sure if the authors accounted for this. Melting of sea ice along the way may also influence the salinity and hence the Freshening Length.

Melting sea ice a one of the contributors to $q$ and hence it is included in our calculation.

**Specific comments**

Title: I find the title misleading and difficult to understand. The study does not investigate *"changes in the surface salinity gradient"*. It should either read *"changes in the sea surface salinity along the current"* or *"the sea surface salinity gradient along the current"* but not *"changes in the gradient"* (which would correspond to the second rather than a first derivative). This mistake is repeated later, for instance in the abstract.

Also, I am not sure what is meant by *"the climate perspective"* in the title.

Title will be modified to it the new focus.

line 4, "surprising": I do not find it surprising that the Irminger Current can be identified based on salinity maxima, given that it represents a saline current system around the fresher subpolar gyre.

The "surprise" is associated with the climatological data i.e. that the salinity maximum is not masked by the massive averaging. This clarification will be added in the text.

line 12: *"A temperature-salinity analysis shows that the Irminger Current east of Greenland is characterized by a compensating isopycnal exchange of temperature and salinity, while west of Greenland the horizontal convergence of less dense surface water is accompanied by downwelling/subduction."*

This sentence is misleading and confusing. It suggests that less dense water is subducted beneath denser water.

The confusion will be clarified in the revised version.

line 79: The SODA data set contains very irregular measurements in time. Thus, it is likely biased towards the recent period.

On periods shorter than the complete 37 year record our analysis does not show any biases in SODA's SSS data. See also our response to general comment #5 above.

line 120: The title and figures refer to the sea surface salinity but the Freshening Length itself seems to be integrated over the full current depth. I find this confusing. To avoid misunderstandings, it would be great if the authors could clarify this in the text and if necessary, adjust the title.

No, the Freshening Length is evaluated solely from surface salinity values.

In case the analysis is restricted to the surface, the Freshening Length would not be a meaningful indicator of the transport fraction that travels around Greenland, as part of the freshwater could be mixed down to depth.

The mixing "down to depth" (i.e. subduction) implies a balancing horizontal convergent flow at the surface. This balancing horizontal flow from the surrounding fresher water is precisely the cause of decrease in salinity of the salty IC

line 178: The conclusion, that the transport loss occurs in eddies is not supported by the preceding sentence. It is not clear from the sentence or the paragraph why the transport loss should occur in eddies. The writing here could be more precise.

The point will not appear in the revision where the transport will be evaluated directly.

line 193: I understand that the Freshening Length is only robust on climatological mean data. However, considering the large interannual and decadal variability in the subpolar region, it is questionable if the application of the Freshening Length to only the climatological average contains meaningful information.

No. As was shown in Berman et al. (2019) the application of the schema to data collected in particular field campaigns yields similar results to that of climatological data provided the data quality is high enough.

line 195: *"...the Freshening Length estimate of the transport is much more robust and informative than direct estimates based on velocity profiles..."*

I strongly disagree with this sentence and the entire paragraph. Of course, the method used always depends on the question that needs to be answered. Still, I would argue that direct estimates based on velocity profiles are generally preferred to calculate transports.

Following the change of focus and the new direct calculations of the transport the entire paragraph will be eliminated from the revision.

Figure 1: in panel b, the labels are difficult to read

Panel b will be deleted from the revised version.

Figure 4: I am not sure why the red and blue lines are fitted to the points. What information is gained from doing this?

Thanks. The graphical issues of Fig. 4 will be corrected in the new version.

---

## Author Comment (AC2)

In this Author Comment #2 the original comments by Reviewer #2 are written in black while the authors' responses are written in blue.

We thank the reviewer for his/her review and for pointing out to us statements that contradict previously published studies. The major and specific comments raised by the reviewer and our detailed response to each of them are listed below. We note that direct estimates of the transports east and west of Greenland will be carried out in the revision which will fundamentally change the paper's main sermon from using $L$ as a proxy for transport ($F$) to using $q=F/L$ as a proxy for the entrainment rate of the surrounding water.

This paper introduced an analysis method, Freshening Length, and applied this to determine the changes of Irminger Current on the both sides of Greenland using SODA reanalysis data.

I have four major concerns/comments for their consideration:

1. Many times the authors stated in the paper that the West Greenland Current (WGC) is a continuation of the Irminger Current. In fact before becoming the WGC, the Irminger Current has merged with East Greenland Current on the east side, referred to as EGC/IC (or EGIC) (e.g., Cuny et al, 2002; Sutherland and Pickart, 2008). I would suggest the authors to improve the introduction with more up-to-date references.

   Sutherland and Pickart, 2008 was referenced in the previous version and the drifter data of Cuny et al. (2002) will be referenced in the revision. The revised version will include a definition of the Irminger Current east of Greenland as the high salinity (S>34.8) current while the EGC is a low salinity (S<34.8) current that flows parallel to the IC along the Greenland shelf while maintaining its low salinity signature. These distinctive characteristics prevail throughout the entire (south-) east coast of Greenland (see Figure 16 of Sutherland and Pickart, 2008). The existence of the EGC will be employed in the revised version to explain the difference in the rates of entrainment of surrounding waters by the IC east and west of Greenland.

2. The author determined the Irminger Current by the maximum salinity at surface. To do that, the authors seem to believe the Irminger Current only contain Irminger Water and this water situates at surface. However, the EGC/IC and WGC contain multiple water masses vertically and the Irminger Water resides away from the surface (~ 500m depth, Pacini et al., 2020). As such, the results shown in the paper can reflect neither the change of Irminger Current nor Irminger Water.

   The various sources that make-up the Irminger Current are not relevant to our sermon. The Current is clearly identifiable by the (local) salinity maximum (S > 34.8, see blue triangles in Figure 2) and the calculations of the Freshening Lengths east and west of Greenland are based solely on the decrease in SSS along it. Both the high SSS of the IC and the freshening (& cooling) along it as it flows around Greenland are its defining characteristics.

3. I am confused by the Freshening Length. The q needs to be better defined. What is the unit of q? m/s? otherwise the equation doesn't make sense as the unit of HU is m^2/s. How to determine the H? The L on the east is 5 times more than the one on

the west. My understanding is that the less L means freshening in a shorter distance or faster freshening in a certain distance, namely the water freshening is 5-time faster on the west than on the east. I am still confused how it can lead to the conclusion that only 20% of Irminger Current contributes to WGC.

The reviewer's concerns in this comment were addressed in Berman et al. (2019) where the underlying theory of Evaporation length was developed. To help the reader grasp the main sermon we will add the following items in the revision: 1) Theoretical details (including a new figure/sketch) that explain the adaptation of the Evaporation Length paradigm; 2) The adaptation of the Evaporation Length to the Freshening Length; 3) A precise definition of $q$ as the parameter controlling lateral mixing; 4) The expression used for evaluating $L$.

4. The authors also stated that the amount of Irminger Current supporting the WGC was previously unknow and they determined that only 20% of Irminger Water rounds Cape Farewell using the Freshening Length. First, the authors should be aware of the studies by Le Bras et al. (2018) and Pacini et al. (2020). The former computed the transport of each boundary current in the east of Greenland using the Overturning in the Subpolar North Atlantic Program mooring data, while the latter reported the transports in the west of Greenland using the multiple-year mooring array on the west side. Their results clearly suggested that the most of the boundary currents in east Greenland flow towards the west Greenland with only a few recirculation at Cape Farewell (interacting with Eirik Ridge). How could the authors explain the discrepancy between the main conclusion of this paper and the compelling observations?

The immediate answer to this dilemma is the different data used in the studies. Our climatological data can be expected to differ from data obtained in particular field campaigns. In the revision the directly calculated values of $F$ east and west of Greenland will be combined with the calculated values of $L$ to yield estimates of the mixing with the surrounding water (which is much more efficient on the west side). This conclusion is consistent with the qualitative results shown in (old) Fig. 3 where the density of the surface water changes on the west side much more than on the east side. This point will be highlighted in the revised version.

Specific comments:

Line 19. should be Faroe Bank Channel.

Thanks. Will be corrected in the revision

Line 20. The connection is not only with Greenland Sea. I would suggest to say Nordic Seas instead.

Will be corrected in the revision

Line 21. EGC doesn't have to be fresh, particularly in the deep layer it also contributes the dense overflow water.

Our statement follows the claims made in de Stuer et al. (2009) and Dodd et al. (2009). The contradicting views on fresh water in the deep part of the EGC are of no consequence to the statement in our work that deals with the near surface salinity signal.

Line 26-27. What currents are they talking about? I don't see the difference. The currents flow continuously northward in Fig. 1a as well.

This point is irrelevant to the new focus of the revised version that will not include Fig.1 .

Figure 1. The two schematics have some discrepancies which add confusions, e.g. EGC and IC are two separate currents in upper panel, while in the bottom panel they are referred as a single merged current – EGIC (green). Why not just show one up-to-date schematic?

Figure 1b will be eliminated from the revised version

Line 80-85. Mark your study region somewhere in Fig.1 or Fig. 2. Label all of the geographic names that were mentioned in the paper, e.g., Cape Farewell.

Will be corrected as required in the revision.

Line 95-100. How could the mean surface hydrography suggest a cyclonic circulation?

The cyclonic circulation is suggested by the climatological means and by prior sketches of the circulation and not by the surface hydrography. The point will be clarified in the revised version.

Line 140. It IS important… and did you mean Fig. 1?

Thanks for detecting the typos which will be corrected in the revised version.

I suggest to change the subtitle to Discussion and Conclusions.

Right. Suggestion will be implemented in the revised version.

Line 155. Downwelling can flux the saltier water towards the coast in the upper layer, while upwelling can transport the fresher water offshore which may influence the water carried by WGC.

Our scenario results from the fast rate of cooling west of Greenland which is consistent with the notion of subduction but the fast freshening rate there requires an explanation. The point will be clarified in the revised version.

Last paragraph. As I said above, the authors should be aware of the published mooring studies. Also the authors can easily check whether their method works out by using the same SODA reanalysis, e.g., compare the volume transports on the both sides of Greenland.

A new table with direct transport estimates from SODA will be added in the revision.